# Photonic Majorana quantum cascade laser with polarization-winding emission

Song Han[1,6], Yunda Chua[1,6], Yongquan Zeng [2] ✉, Bofeng Zhu[1,3], Chongwu Wang[1], Bo Qiang[1], Yuhao Jin[1], Qian Wang [4], Lianhe Li [5], Alexander Giles Davies [5], Edmund Harold Linfield [5], Yidong Chong [3], Baile Zhang [3] & Qi Jie Wang[1,3] ✉

Topological cavities, whose modes are protected against perturbations, are promising candidates for novel semiconductor laser devices. To date, there have been several demonstrations of topological lasers (TLs) exhibiting robust lasing modes. The possibility of achieving nontrivial beam profiles in TLs has recently been explored in the form of vortex wavefront emissions enabled by a structured optical pump or strong magnetic field, which are inconvenient for device applications. Electrically pumped TLs, by contrast, have attracted attention for their compact footprint and easy on-chip integration with photonic circuits. Here, we experimentally demonstrate an electrically pumped TL based on photonic analogue of a Majorana zero mode (MZM), implemented monolithically on a quantum cascade chip. We show that the MZM emits a cylindrical vector (CV) beam, with a topologically nontrivial polarization profile from a terahertz (THz) semiconductor laser.

In recent years, the concepts underlying topological quantum materials[1] have inspired a variety of novel topological photonic devices[2-27]. For example, topological lasers (TLs) have been realized in a variety of designs, including 1D Su-Schrieffer-Heeger (SSH) lasers with localized topological modes[20-23], 2D SSH lasers with corner modes[24-27], and 2D photonic crystal and coupled-resonator lasers with chiral edge modes[1-8,15-19]. One of the most promising features of TLs is the insensitivity of their lasing modes to certain perturbations, which may reduce the impact of fabrication defects or environmental disturbances. Very recently, researchers have started to explore using TLs to form nontrivial emission patterns, such as vortex beams[7,8]; in such lasers, the topological properties of the internal photonic modes determine the far-field features of the emitted light, including its topological structure. In ref. [7], a photonic spin Hall insulator was optically pumped with a spatially tailored beam to generate a vortex beam carrying out-of-plane orbital angular momentum (OAM); and in ref. [8], a laser with a high-order OAM beam was achieved using a

photonic quantum Hall lattice biased by a strong external magnetic field. These approaches require either structured optical pumping or an external magnetic field. It is desirable to develop an electrically pumped TL that can convert electrical energy directly to a laser beam with nontrivial structure.

Here, we demonstrate an electrically pumped TL based on a Majorana zero mode (MZM), possessing single-mode lasing and a nontrivial polarization-winding emission profile corresponding to a cylindrical vector (CV) beam. The photonic MZM is a spectrally isolated mid-gap state occurring in a photonic structure formed by a honeycomb lattice with chiral Kekulé phase modulation[28-35]. The modes of the system are described by a 2D Dirac equation with mass vortex, and as originally shown by Jackiw and Rossi, there must be a zero-energy (i.e., mid-gap) solution which is topologically protected by the winding of the mass vortex[28,29]. In the photonic context, this MZM is useful for lasing because its frequency is pinned to the center of the photonic bandgap, and also because, as further discussed below, its

[1]Centre for OptoElectronics and Biophotonics, School of Electrical and Electronic Engineering & The Photonics Institute, Nanyang Technological University, Singapore, Singapore. [2]Electronic Information School, Wuhan University, Wuhan, China. [3]Division of Physics and Applied Physics, School of Physical and Mathematical Sciences, Nanyang Technological University, Singapore, Singapore. [4]Institute of Materials Research and Engineering, Agency for Science, Technology and Research (A*STAR), 2 Fusionopolis Way, #08-03, Innovis, Singapore 138634, Singapore. [5]School of Electronic and Electrical Engineering, University of Leeds, Leeds, UK. [6]These authors contributed equally: Song Han, Yunda Chua. ✉e-mail: yqzeng@whu.edu.cn; qjwang@ntu.edu.sg

intrinsic chirality generates a nontrivial far-field emission pattern[34]. The TL cavity is monolithically implemented on a quantum cascade laser (QCL) wafer, based on intersubband electron transitions within multiple quantum wells. Unlike previous TLs which required careful tailoring of the pumping region to avoid unwanted lasing modes[1–8], the present device is electrically pumped using simple top and bottom metal contacts covering the entire chip. The mid-gap Majorana laser mode is identified by spectral scanning over the full dynamic range of the laser, and far-field measurements reveal a doughnut-shaped laser beam with a polarization singularity at the beam center, characteristic of a CV beam. Different from the existing methods for generating CVBs that rely on bulky free-space optical components[36], the CVB generator developed in this work is a monolithic semiconductor laser at an ultracompact footprint. The laser delivers a highest output power larger than 9 mW that is 5.9 times of the ridge laser with similar size. This compact and efficient laser, with at-source CV beam profile, has potential applications for THz lidar, imaging, microscopy, and wireless communications.

## Results

### Theoretical model and sample fabrications

The topological cavity hosting the photonic MZM is implemented by an air-hole hexagonal lattice drilled through the top contact metal layer and active medium of a THz QCL wafer (Fig. 1a). Since the photonic lattice is cladded by double metal layers, it supports transverse magnetic (TM) polarized modes. The pump current is supplied by a wire bonding pad insulated by a thick $SiO_2$ layer, which can be seen in the scanning electron microscope (SEM) image in Fig. 1b. In the pristine lattice (with lattice constant $a$, side length $d = a/\sqrt{3}$, and air holes having uniform radius $R_0 = 0.35d$), the photonic crystal exhibits doubly degenerate Dirac cones, or "valleys", at wavevectors $K_{\pm} = [\pm 4\pi/3d, 0]$ and frequency $\omega_D$ (Fig. 2a). A vortex-like Kekulé modulation is overlaid on the lattice, in the form of position-dependent air-hole radii obeying

$$R(r) = R_0 + \Delta R(r)\cos[K \cdot r + \theta(r)], \quad (1)$$

where $K = K_+ - K_-$ implies the intervalley coupling, $r = (x, y)$ is the position vector in Cartesian coordinates, $\Delta R(r) = \Delta R \tanh(r/\xi)$ is a

radial profile, $\xi = 2a$ is the vortex core radius, and $\theta(r) = w\tan^{-1}(y/x)$ is a position-dependent phase factor with winding number $w = +1$, as shown in Fig. 2b. The resulting variation in the air-hole radii is plotted in Fig. 2c. The Kekulé modulation induces intervalley coupling and opens a bandgap around $\omega_D$ of around 12% bandwidth (Fig. 2a). The Jackiw-Rossi binding mechanism[28–35] generates a photonic MZM that is tightly localized to the vortex core, as verified by numerical simulations (Fig. 2d). The breaking of the photonic crystal's inversion symmetry (from $C_{6v}$ to $C_{3v}$) also results in radiative coupling to the out-of-plane continuum, and numerical simulations reveal that the photonic Majorana mode has a doughnut-shaped intensity profile in the far field (Fig. 2e).

### Experimental results

The THz QCL wafer supplies gain over the 2.9 THz to 3.8 THz range, which overlaps with the designed photonic bandgap (see Supplementary Materials). Two different devices with same lattice constant $a = 30\ \mu m$ but different winding numbers, i.e., $\pm 1$, were fabricated. Their measured emission spectra at various pumping current densities are plotted in Figs. 3a and 3b, respectively. With increasing pump, each emission spectral envelope undergoes a gradual blueshift, due to the Stark shift of the intersubband transition in the THz QCL medium[37]. We can nonetheless clearly identify the MZM peaks, which lie at around 3.47 THz for both samples, very close to the predicted mid-gap frequency. We also observe a weaker emission peak at 3.63 THz for the two respective samples; these are identified as upper band edge (UBE) modes since they occur at the upper edge of the bandgap as predicted by numerical calculations. These experimental results are also consistent with numerical calculations of the modal net gain coefficients (Fig. 3c); see Supplementary Materials for details about these calculations. The lower band edge (LBE) modes, which are predicted to have the lowest net gain coefficients, thereby do not appear in the experimental emission spectra. From the experimental light-current-voltage curves (Fig. 3d), the electrical pumping dynamic range for laser emission can be observed with clear lasing threshold around 1.2 kA/$cm^2$ and output power roll-over point at 1.45 kA/$cm^2$ for both samples. The intensity of the main peak corresponding to MZM is around 30 times higher than the strongest UBE peak. Therefore, the SMSR is over 14.7 dB, which is possibly aided by the UBE frequency being

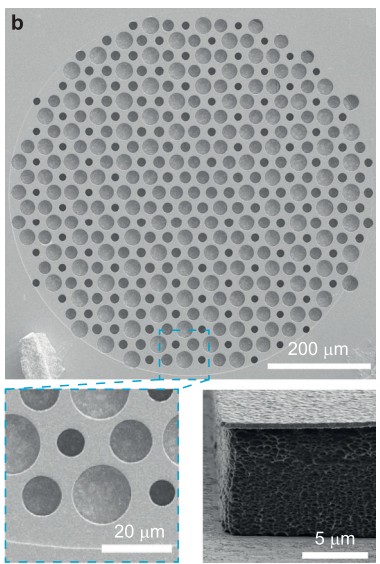

**Fig. 1 | Electrically pumped quantum cascade laser (QCL) based on a photonic Majorana zero mode (MZM). a** An air-hole structure is patterned into the top Au/ QCL layers. The top and bottom Au layers work as electric conductors for pulsed pump current injection, as well as tight mode confinement in the vertical direction due to the small Ohmic losses at terahertz frequencies. The region outside the

cavity is insulated by a $SiO_2$ layer so that it is unpumped. **b** SEM image of a fabricated sample, which has lattice constant $a = 30\ \mu m$, winding number $w = +1$, vortex core radius $\xi = 2a$, and an overall device radius of six periods. The side view of the SEM shows that the QCL is undercut relative to the top Au contact.

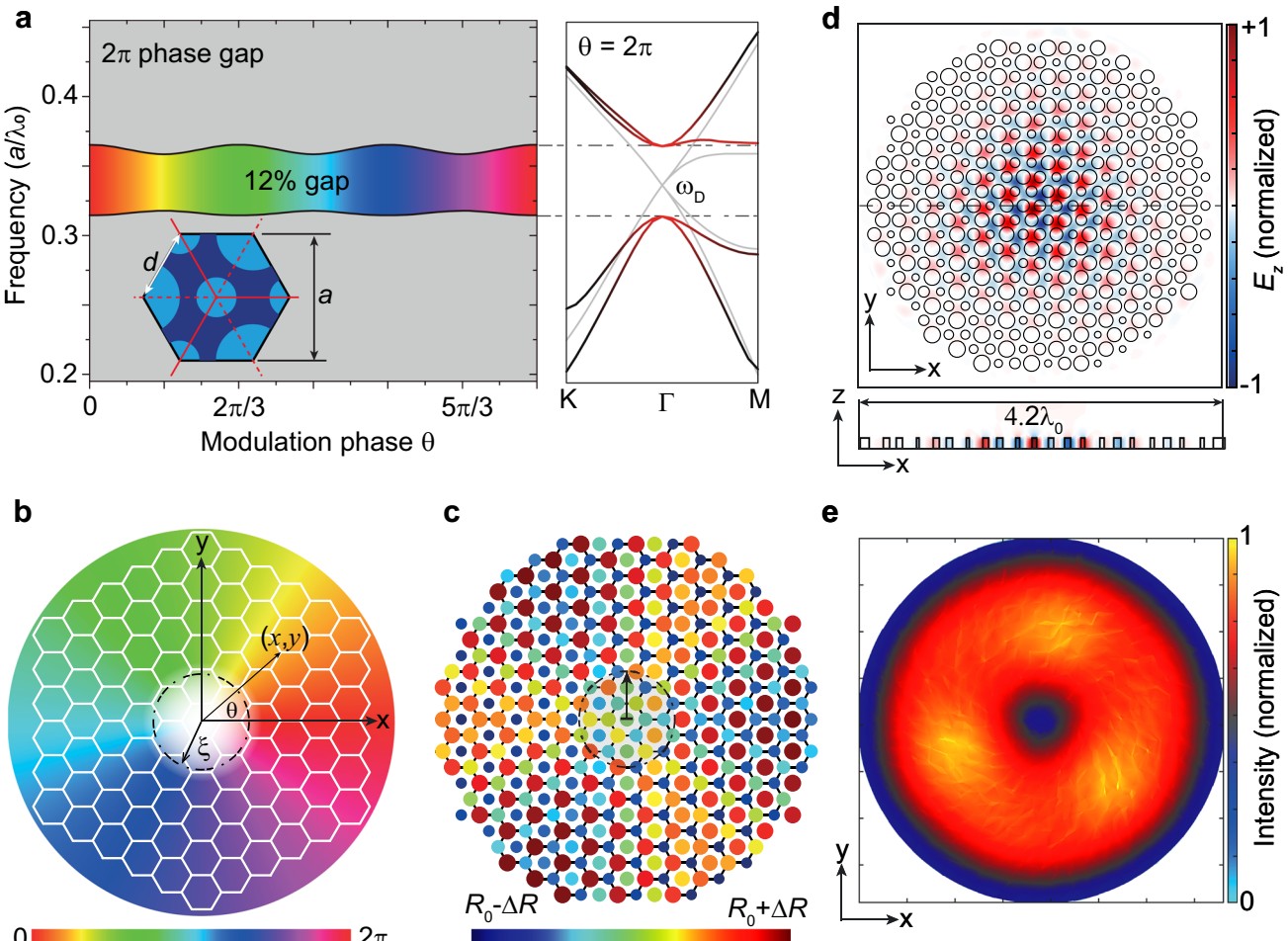

**Fig. 2 | Numerical simulations of the topological cavity. a** Band diagram of a hexagonal supercell with a Kekulé modulation (left panel) and the unmodulated photonic crystal (right panel). The bandgap has ≥12% relative width. Inset: schematic of the hexagonal supercell and its Brillouin zone. **b** The phase of the Kekulé pattern, which has winding number of +1. The colorbar defines the smooth phase varying from 0 to 2π. **c** Modulation of the photonic structure's air-hole radii, using a vortex core radius of two periods ($\xi = 2a$). The colorbar defines the smooth radii varying from $R_0 - \triangle R$ to $R_0 + \triangle R$, here $R_0 = 0.35a$ and $\triangle R = 0.15a$, respectively. **d** Calculated electric field ($E_z$) for the photonic MZM, which is tightly confinement to the vortex core. **e** Calculated far-field intensity pattern of the photonic MZM, which exhibits a doughnut-like profile characteristic of cylindrical vector (CV) beams.

located in the tail of the gain region (Fig. 3c). The measured peak power is around ~1 mW for the laser devices under investigated, which is mainly limited by the power performance of QCL wafer (see Supplementary Materials for more details). The laser emission at different operating temperatures were also characterized. The Majorana lasing mode always dominates the emission spectra and pined at an invariant lasing frequency if the slight blueshift induced by thermal effect is not considered. The SMSR gradually decreases from 14.7 dB at 8.5 K to 10.6 dB at 40 K, which is common for THz QCLs as peak gain coefficient decreases during temperature rising (see Supplementary Materials for details).

The far-field beam profile was probed using a custom-made intensity scanner apparatus, shown in Fig. 4a. The diverged laser beam is collimated by a lens with focal length 5 cm, and collected by a THz Golay cell detector. Numerical calculations predict that the Majorana mode produces a CV beam with a doughnut-shaped far-field intensity profile (Fig. 4b), and our experimentally obtained intensity profile is in good agreement (Fig. 4c). The far-field beam profiles were also measured at different operating temperatures, which show similar doughnuts from 8.5 K to 40 K even the SMSR decreases (see Supplementary Materials). Such robust emission profiles, in some sense, demonstrate the topological robustness of the Majorana-like photonic QCLs.

Aside from the intensity profile, the electric field vector of the CV beam winds by 2π around the vortex core (Fig. 4b), forming a far-field nontrivial topology $w = +1$. This can be observed by using a linear polarizer to filter the cross-polarized components during the far-field intensity scanning. Numerical calculations show that this method divides the far-field beam into two lobes with orientations slightly deviating from the polarizer direction (Fig. 4d and e, upper right panels). In our apparatus, a wire-grid-based linear polarizer with one-inch aperture was inserted between the collimating lens and the Golay cell detector. With the linear polarizer rotated by 45° and 135°, we obtained far-field profiles with two lobes arranged along $x$ and $y$ axis respectively, very similar to the calculation results (Fig. 4d and e, lower right panels). This validates the far-field polarization-winding properties of Majorana-like photonic QCLs.

**High-performance THz topological MZM QCL**

We also fabricated a laser device with larger size to obtain high-power emission. The geometric parameters of the device are $a = 30$ μm, the winding number $w = -1$, the vortex core radius $\xi = 8a$ (i.e., 240 μm), and the radius of total device is $17a$ (i.e., 510 μm). With such large-area laser, the output intensity is increased significantly, as shown by the L-I-V cures in Fig. 5a. The

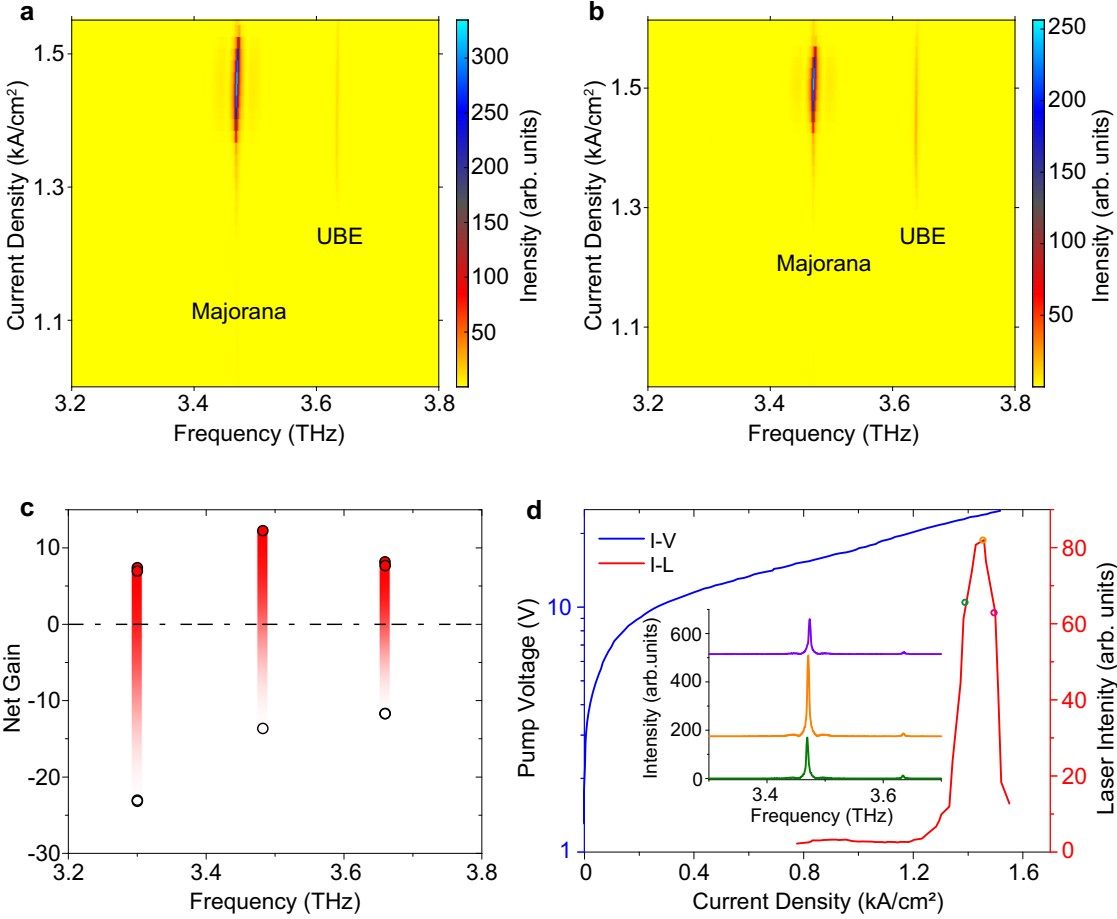

**Fig. 3 | Observation of lasing Majorana zero modes. a, b** Emission intensity spectra for different pump current densities, obtained using samples with lattice constant $a = 30\,\mu m$ and winding number of ±1, respectively. **c** Calculated net gain coefficients before and after the laser devices are pumped. **d** Light-current-voltage (L-I-V) curve for the sample with winding number of +1 (the other sample behaves similarly, see the Supplementary Materials for more details). These intensities are obtained by integrating over only the emission peak of the photonic MZM. The inset figure shows typical emission spectra at different pump current intensities, from which the maximum SMSR is calculated to be larger than 14.7 dB.

maximum peak power is estimated to be 9.04 mW, which is around six-fold higher than that of a ridge laser from the same fabrication process (see Supplementary Materials for details). Compared to the other photonic MZM laser (vortex core radius of $\xi = 2a$), the enhancement of power emission is about eight-fold for such large-area device (vortex core radius of $\xi = 8a$). A stable single-mode laser mode can be observed when continuously increasing the pump current densities, as shown in Fig. 5b. The SMSR estimated from a spectrum with the maximum output power is larger than 15 dB, as shown in Fig. 5c. Therefore, the emission power of topological MZM QCL can be further improved by simply scaling the cavity size while maintaining single-mode operation and CVB emission with nontrivial polarization winding.

## Discussion

The CV beams, with their doughnut-shaped beam profiles that can give rise to tightly focused spot, have found revelational breakthroughs in super-resolved fluorescence microscopy[38-40]. They also carry different polarization windings, such as radial polarization, azimuthal polarization, and spiral polarization (the current study shows such a polarization), that can be applied to different types of super-resolved microscopies, optical trappings and manipulations, laser machining, and charged particle acceleration. These advanced applications have been reviewed by ref. [36]. The realization of THz CV beams through a monolithic approach can find its significance for applications in high-bandwidth wireless communications, THz

security inspection, THz spectroscopy, bio-imaging, etc. The demonstrated THz CVB source is monolithically integrated and electrically pumped, which can be easily integrated onto photonic chips for applications listed above. Last but not least, our laser cavity design only relies on dielectric refractive index modulation. Therefore, it can be easily scaled to other wavelength regimes, such as the mid-IR, the near-IR, and the visible regions.

In summary, we have demonstrated electrically pumped THz lasers based on photonic MZMs, which generate cylindrical vector beams with nonzero polarization winding. The laser emission is dominated by a mid-gap MZM, with a SMSR of >14 dB. The winding of the CV beam is intrinsically tied to the vorticity of the Kekulé modulation on a photonic lattice, demonstrating how 2D topological modes can affect the topological features of the laser emission in the far field. It is worth mentioning that the laser cavity design in this work enables the control of emission spectrum and far-field beam polarizations simultaneously, making Majorana-like photonic QCLs unique as compared to standard THz QCLs, such as ridge laser, distributed-feedback laser, and traditional photonic crystal laser which have no such flexibility. The emission properties such as output powers, beam divergences, polarizations can be further optimized through tailoring the structural parameters such as air-hole radius and cavity size. While preparing this manuscript, the authors noticed two independent works demonstrating an optically pumped near-infrared laser based on photonic MZMs[41,42].

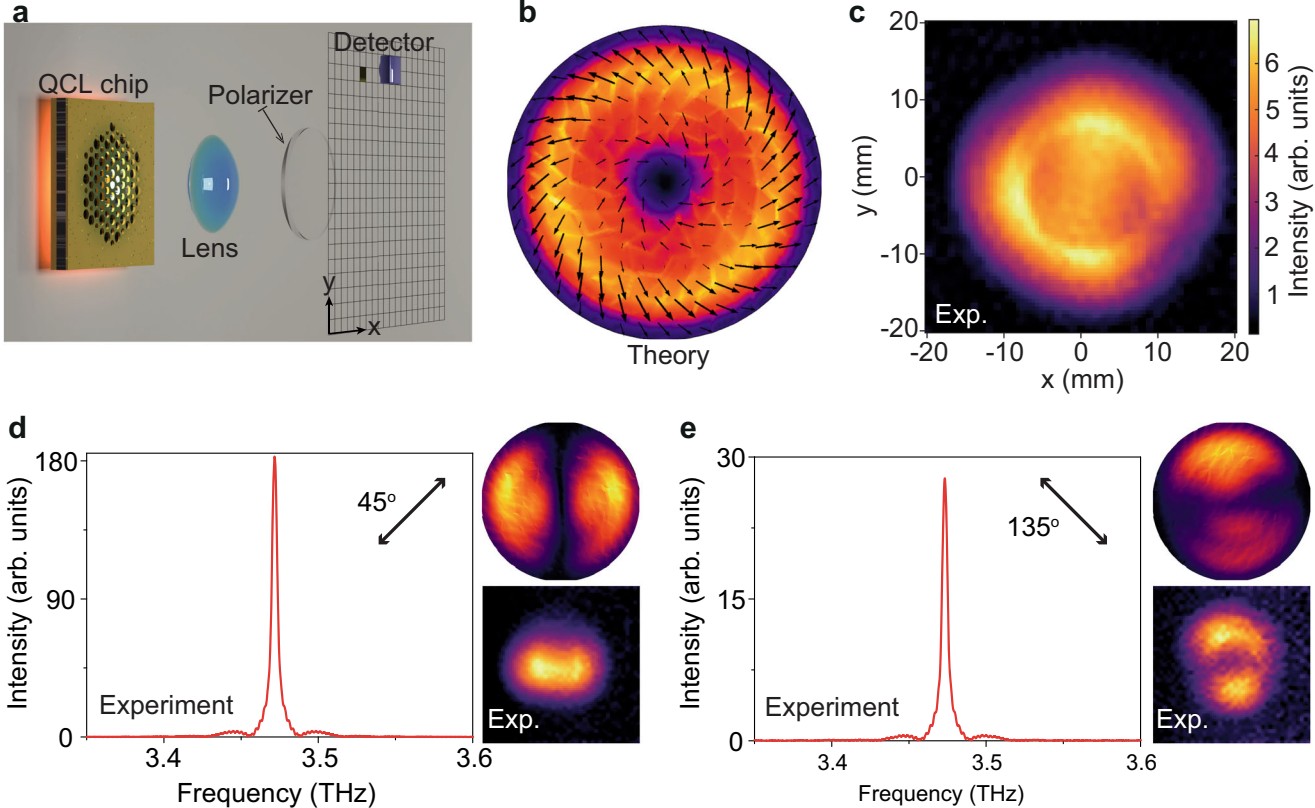

**Fig. 4 | Far-field characterization of the CV beam. a** Experimental setup for the far-field beam measurements. The diverged CV beam is collimated by a focal lens, and probed by a Golay cell THz detector mounted on a mechanical stage. **b, c** Numerically calculated and experimentally obtained intensity profiles.

**d, e** Results obtained by inserting a wire-grid polarizer between the focal lens and THz detector, with polarization angles of 45° and 135° respectively. Left panels: polarization-resolved emission spectra. Right panels: numerically calculated (upper) and experimentally measured (lower) intensity patterns.

## Methods

### Device fabrication

A THz QCL wafer with a three-well resonant-phonon GaAs/$Al_{0.15}Ga_{0.85}As$ design (see Supplementary Materials Fig. S1 for detail) was used in this work. The gain curve spans from 2.9 THz to 3.8 THz, verified by the emission spectrum envelope of a ridge laser fabricated on the same wafer. The topological cavities were patterned onto the wafer using the standard metal-semiconductor-metal configuration (Fig. 1a). The fabrication process began with metal (Ti/Au, 20/700 nm) deposition by an electron-beam evaporator onto the THz QCL wafer and an n⁺-doped GaAs host substrate, followed by Au/Au thermo-compression wafer bonding. Wafer polishing and selective wet etching using $NH_3 \cdot H_2O/H_2O_2/H_2O$ (3/57/120 ml) solution were sequentially conducted to remove the THz QCL substrate down to an etch-stop layer. The etch-stop layer was removed by 49% hydrofluoric acid solution, and the QCL active region was exposed for subsequent microfabrication. A 300-nm $SiO_2$ insulation layer was deposited onto the THz QCL wafer using plasma-enhanced chemical vapor deposition, followed by optical lithography and reactive-ion etching (RIE) to define the pumping area. The photonic structure patterns were transferred onto the THz QCL wafer by optical lithography, with deposition and lift-off to define the top metal or electrode layer (Ti/Au/Ti/SiO₂, 20/300/20/400 nm), with the $SiO_2$ layer used as a hard mask for the electrode during the etching of active region. With the top metal layer as a hard mask, the photonic structures were formed by reactive-ion dry etching through the active region with a gas mixture of $BCl_3/CH_4/Cl = 100/20/5$ standard cubic centimeters per minute. The top metal layer (remnant thickness approximately 300 nm) was retained as a top contact for current injection. The host substrate was covered by a Ti/Au (15/200 nm) layer as bottom contact. The SEM was used to capture the etching side wall image and calibrate the air-hole size. Finally, the THz QCL chip was cleaved into small pieces, indium-soldered onto a copper heatsink, wire-bonded and attached to a cryostat cold finger for characterization.

### Characterization

For the emission spectrum measurements, the fabricated QCLs were mounted in a helium-gap-steam cryostat with temperature stabilized at 8.5 K and driven by an electrical pulse generator with repetition rate of 10 kHz and pulse width of 500 ns. The spectra were captured by a Fourier transform infrared spectrometer (FTIR, Bruker Vertex 80 series) with a room-temperature deuterated triglycine sulfate (DTGS) detector. The spectral resolution is 0.08 cm⁻¹. A scanning setup was employed for far-field beam profile characterization. Far-field intensity measurements were performed with a THz Golay cell detector (TYDEX GC-1T, collection aperture size is 11 mm) mounted on a 15 cm arm. Before the measurement, a home-developed alignment technique was used to align the laser, collimating lens, and the detector based on the principle of light diffraction (see Supplementary Materials for details). To improve the signal-to-noise ratio, the laser device was modulated by a 15 Hz electrical square signal for lock-in amplification of the detector signal. To further analyze the beam profile, a THz wire-grid polarizer was inserted between the focal lens and Golay cell detector. By continuously rotating the polarizer, both laser spectra and far-field beam profiles were captured.

### Numerical method

All 3D full-wave simulations were conducted by the finite-element method-based software COMSOL Multiphysics. In the simulation, the 10-μm-thick QCL active region was treated as a lossless and dispersion-

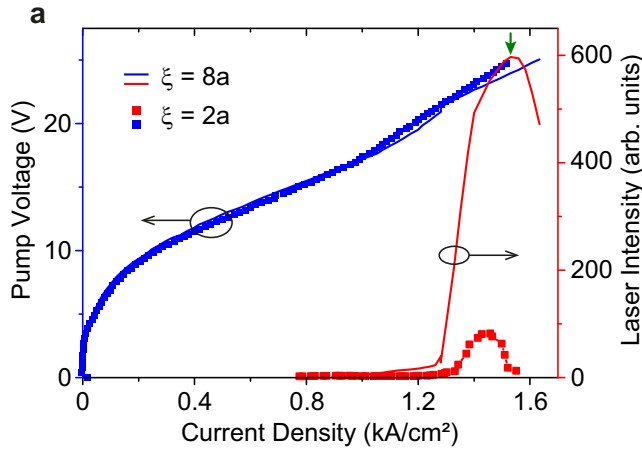

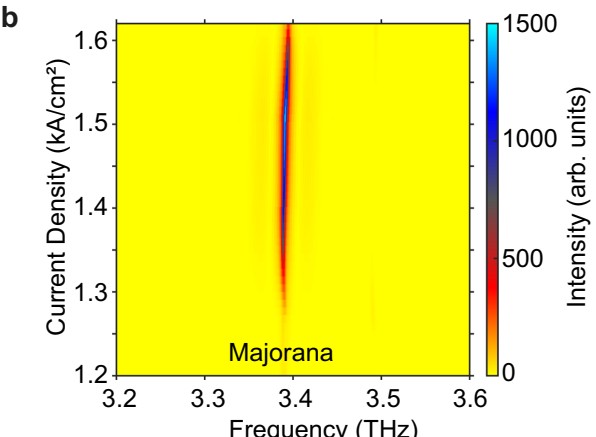

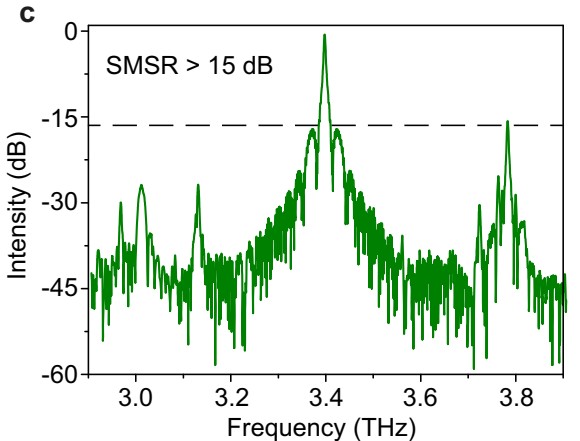

**Fig. 5 | Power scaling of the THz topological MZM laser with a larger size.**
**a** Measure L-I-V cures of topological MZM lasers with vortex core radius $\xi = 8a$ and $\xi = 2a$, the light emission is enhanced about 8-fold for the large-area device ($\xi = 8a$). **b** 2D spectrum map of the laser at different pump current densities. A stable lasing peak is clearly observed at around 3.4 THz. **c** The lasing spectrum obtained at its roll-over point (where the output power is the maximum), showing a SMSR > 15 dB.

free medium with an estimated refractive index of 3.85. Two gold layers forming the top and bottom contacts have a thickness of 600 nm, modeled as lossy metal with refractive index 182.67 + 212.11i. The air-hole pattern was generated with the Layout Editor Software.

## Reporting summary

Further information on research design is available in the Nature Portfolio Reporting Summary linked to this article.

## Data availability

Data supporting key conclusions of this work are included within the article and Supplementary information. All raw data used in the current study are available from the corresponding author under reasonable request.

## Code availability

The codes that support the plots within this paper are available from the corresponding authors upon reasonable request.

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

## Acknowledgements

This work is supported by funding from the Singapore Ministry of Education (MOE), grants MOE-T2EP50120-0009, A*STAR Programmatic Funds (A18A7b0058), and the National Research Foundation Competitive Research Program (NRF-CRP18-2017-02 and NRF-CRP23-2019-0007). Y.C. and B.Z. acknowledge support from the Singapore MOE Academic Research Fund Tier 2, grants MOE2015-T2-2-008 and MOE2018-T2-1-022 (S), and the Singapore MOE Academic Research Fund Tier 3 grant MOE2016-T3-1-006. L.L., A.G.D., and E.H.L. acknowledge the support of the EPSRC (UK) HyperTerahertz programme (EP/P021859/1), and the Royal Society and the Wolfson Foundation. The authors also acknowledge Dr. Penglin Gao and Dr. Di Bao for helpful discussions.

## Author contributions

S.H. conceived the idea of the research, performed the theoretical and numerical calculations. B.F.Z. and B.Q. helped with layout of hexagonal lattice in MATLAB; Y.D.C. carried on the sample fabrications under guidance of Y.Q.Z.; B.Q. and C.W.W. helped with the FIB and SEM photos; L.L., A.G.D., and E.H.L. performed QCL wafer growth; S.H. and Y.D.C. carried out the QCL spectra measurements and the far-field measurement; Q.W. facilitated wafer bonding for device fabrication; Y.J. performed the gain profile calculations. S.H., Y.Q.Z., Y.C., B.Z., and Q.J.W. wrote the paper; all the authors contributed insight and discussion on the results; Q.J.W. supervised the project.

## Competing interests

The authors declare no competing interests.
