## [Peer review file · Nature Communications]

REVIEWER COMMENTS

Reviewer #1 (Remarks to the Author):

The manuscript "Photonic Majorana Quantum Cascade Laser with Polarization-Winding Emission" reports the development of a topological THz QCL. The idea is nice and the presented simulation part very rigorous. However, I do not really see a real breakthrough in the demonstrated device. What are the performances that makes it unique or intriguing? How it can impact THz photonics or optoelectronics?

A few remarks below:

- How the far field appears without collimation lens? What's the beam divergence?
- The power output is rather low. What's the maximum peak power achieved? Which kind of practical use those THz QCLs can have?
- The technological impact of those lasers is not discussed at all. What's the advantage they can have over standard THz QCLs?
- how can the author exclude contribution coming from disordered modes, like the ones arising in a random laser? Can the authors comment on that?

The manuscript appears very technical, not conceived for a broad audience but addressed only to researchers working in the field.

For the above reasons I do not recommend publication in Nature Comm. I suggest submitting it to a different journal or transferring to Scientific Reports.

Reviewer #2 (Remarks to the Author):

The authors demonstrated an electrically pumped terahertz topological laser. The laser emission is dominated by mid-gap Majorana zero modes, which form cylindrical vector (CV) beams with nonzero polarization winding. The winding of the CV beam is intrinsically tied to the vorticity of the Kekulé modulation on the photonic lattice, which was achieved by creating air holes through the semiconductor active region and top metal contact. The device exhibit good performance with peak power over 1 mW, threshold current density of around 1.6 kA/cm², and side-mode-suppression-ratio (SMSR) of more than 16 dB over a single weak high band edge mode. The experimental data is well explained and confirmed by the theoretical simulation, demonstrating the first photonic Majorana quantum cascade laser, which can have important applications in wireless communications, terahertz microscopy and imaging. The conclusions are supported by the experimental and theoretical results. All Figures are well plotted and the manuscript (including the supplementary materials section) is well drafted. I would recommend to publish this paper.

A few optional comments:

- (1) For those theoretical simulations/experimental measurements in Figures 3 and 4, and Figure S8 in the sections 5 and 6 and Figure S12 in the supplementary materials, at what temperature was the device simulated/measured?
- (2) Has the SMSR been measured at elevated temperatures? Does the SMSR depend on device operating temperature? Does the far-field patterns of the CV beam depend on device operating temperature?
- (3) The Figure S8 in the section 6 of the supplementary materials should be Figure S9. The onward figure numbers should be changed accordingly.
- (4) Reference 30, the title should be “Photonic Topological Mode Bound to a Vortex”, and the year should be (2020).
- (5) If possible, can the authors include the active region design information of the QCL in the supplementary materials section?

Response Letter to Reviewers

We are grateful for the detailed and constructive comments and suggestions on this manuscript (NCOMMS-22-05649) from all the reviewers.

In the response letter below, each reviewer comment is quoted in italics and is followed by the corresponding detailed response. In the text below, changes to the revised manuscript as a response to the reviewers' comments are highlighted in blue color and replies/clarifications regarding the reviewer's comments are also provided accordingly.

GENERAL COMMENTS FROM REVIEWER #1:

The manuscript "Photonic Majorana Quantum Cascade Laser with Polarization-Winding Emission" reports the development of a topological THz QCL. The idea is nice and the presented simulation part very rigorous. However, I do not really see a real breakthrough in the demonstrated device. What are the performances that makes it unique or intriguing? How it can impact THz photonics or optoelectronics?

Response from Authors:

We thank the reviewer for the highly positive comments that “*The idea is nice and the presented simulation part very rigorous*”. We also thank the reviewer’s for the concern raised on the impact of the work, which helps us to improve the quality and clarity of the presentation in the revised manuscript.

The main concern of the reviewer is a concrete one: “*What are the performances that makes it unique or intriguing?*” and “*How it can impact THz photonics or optoelectronics?*”. We would like to clarify this from the following two aspects:

First of all, terahertz (THz) vector beams have important applications, such as in THz imaging, communications, spectroscopy, astronomy, and charged particle acceleration. For example, compared to a common Gaussian beam, a vector beam can produce a tightly focused spot (*Physical Review Letters*, 2003, 91(23): 233901), and thus is much desirable for high-resolution imaging (*Applied Optics*, 2006, 45(3): 470-479). Furthermore, radially polarized beams can efficiently excite surface plasmon modes on a metal wire, which is promising for THz communications (*Nature*, 2004, 432(7015): 376-379; *Scientific Reports*, 2016, 6(1): 1-8). Recently, researchers also found that THz vector beam can exhibit advantageous dichroism over conventional circular polarization beam in the spectroscopy of magnetic excitations (*Physical Review Letters*, 2019, 122(23): 237401; *Physical Review Letters*, 2021, 126(15): 157401). In addition, as frequencies of many astrophysical sources are in the THz range, the THz vector beams can also be used for applications in astronomy, such as rotating black holes detection (*The Astrophysical Journal*, 2003, 597(2): 1266; *Astronomy & Astrophysics*, 2008, 488(3): 1159-1165; *Nature Physics*, 2011, 7(3): 195-197).

So far, THz vector beams are in general generated by employing free space bulk optics such as segmented half-wave plate (*Optics Letters*, 2017, 42(1): 41-44), dielectric metasurfaces (*Nanophotonics* 2019; 8(7): 1263-1270; *Nanophotonics* 2020; 9(10): 3393-3402), photoconductive antennas with concentric electrodes (*Applied Physics Letters*, 2005, 86(16): 161904), a nonlinear crystal axicon illuminated with a circularly polarized femtosecond laser

beam (*Applied Physics Letters*, 2021, 119(22): 221110), and segmented nonlinear crystals with rotated crystal axis (*Optics Express*, 2012, 20(20): 21896-21904). However, these methods require external THz sources and free-space optics, and therefore, the consequent systems must be bulky. To the best of our knowledge, **the generation of THz vector beams based on a monolithic and integrated semiconductor laser chip has not been previously reported.**

In this work, we have, for the first time, generated a THz vector laser beam from a compact chip directly. Scientifically, it is achieved through a topological photonic design, i.e. a photonic analogue of Majorana zero mode. Technologically, it realizes not only a THz vector beam emission, but also a single mode lasing operation, in the same topological photonic cavity. The polarization and the lasing frequency of the vector beam can be further controlled by the topological lattice parameters without sacrificing the single-mode operation. Therefore, we believe this is a breakthrough in both fundamental science and practical technology, allowing us to achieve integrated and miniaturized THz devices for the various applications mentioned above.

We would like to further elaborate on the unique capability of our device in controlling the emission spectrum and beam polarization at the same time. Conventionally, these two properties need to be controlled separately. For example, one-dimensional distributed feedback gratings and two-dimensional photonic crystals (PhCs) have been widely used to realize single mode emission, but they lack the far-field tailoring freedom in achieving complex far-field beam properties such as the vector beams. Another example is the plasmonic slits that have been created on semiconductor emission facet and/or an external waveguide to control the emission polarization (*N. Yu, et al, APL 2009, 94, 151101*; *G. Liang, et al, ACS Photon. 2017, 4, 517-524*), but the laser cavity and the plasmonic slits need to be separated in order to control the emission spectrum and beam polarization independently. For the cases where the plasmonic structures are integrated within the laser cavity, e.g., plasmonic gratings on top of the ridge laser top contact, (*P. Rauter et al, PNAS 2014, 1421991112*), the lasers are operated in multimode emission and the far-field pattern are chaotic. This is because the plasmonic structures will inversely affect the lasing cavity (the phase and amplitude of the lasing mode) in an uncontrolled way thus destroying the lasing performance of the laser once those plasmonic structures are integrated within the laser cavity.

In contrast, our device can control both emission spectrum and beam polarization at the same time. The single-mode operation in our device is guaranteed by the topological photonic Majorana zero mode, i.e., the single lasing mode always appears in the midgap of the bandgap. The polarization control, or polarization winding in the far field, stems from the topological charge through Kekule phase modulation around the vortex core. These are the unique and intriguing properties which cannot be achieved through traditional photonic designs for semiconductor lasers.

In the revised manuscript, we have emphasized the significance of single-mode THz vector beam generator and its applications on pages 5 and 6 in the “Discussion and Conclusion” part, which reads:

“The CV beams, with their doughnut-shaped beam profiles that can give rise to tightly focused spot, have found revolutionary breakthroughs in super-resolved fluorescence microscopy [37-39]. They also carry different polarizations, such as radial polarization, azimuthal polarization, and spiral polarization (the current study shows such a polarization), that can be applied to different types of super-resolved microscopies, optical trapping and manipulation, laser machining, and charged particle acceleration. These advanced applications have been reviewed by Ref. 40. THz wave

emission devices are least developed, compared to the wave generators in the adjacent infrared and microwave frequency ranges. The realization of THz CV beams through the proposed integrated approach can find its significance for applications in high-bandwidth wireless communications, THz security inspection, THz spectroscopy, bio-imaging, etc. The demonstrated THz CVB source is monolithically integrated and electrically pumped, which can be easily integrated onto photonic chips for applications listed above. Last but not least, our laser cavity design only relies on the dielectric refractive index modulation. Therefore, it can be easily scaled to other wavelength regimes, such as the mid-IR, the near-IR, and the visible regions.

In summary, we have demonstrated the first electrically pumped THz lasers based on photonic Majorana zero modes, which generate cylindrical vector beams with nonzero polarization winding. The laser emission is dominated by a mid-gap Majorana zero mode, with a SMSR of >14 dB. The winding of the CV beam is intrinsically tied to the vorticity of the Kekulé modulation on a photonic lattice, demonstrating how 2D topological modes can affect the topological features of the laser emission in the far field. It is worth mentioning that the laser cavity design in this work enables control of emission spectrum and far-field beam polarizations simultaneously, making Majorana-like photonic QCLs unique as compared to standard THz QCLs, such as ridge laser, distributed-feedback laser, and traditional photonic crystal laser which have no such flexibility. The emission properties such as output powers, beam divergence, polarizations can be further optimized through tailoring the structural parameters such as air hole radius and cavity size.”

SPECIFIC COMMENTS FROM REVIEWER #1:

Reviewer #1 -- Comment 1:

How the far field appears without collimation lens? What's the beam divergence?

Response from Authors:

We thank the reviewer for this comment.

To illustrate the far-field beam pattern and its divergence, we've done a 3D full-structure simulation in COMSOL, as shown in **Figure R1** below. Without lens collimation, the far field profile shows a typical Lagurre-Guass beam profile, as shown in **Figure R1(a)**. To investigate the divergence angle, a 2D plane view at $y = 0$ plane is visualized, as shown in **Figure R1(b)**. The divergence angle is estimated to be around 30° . In the measurement, a collimation lens with focal length of 5 cm was employed to convert the diverging beam to be a collimated beam. Such a method can save us scanner space, and more importantly, make it easier to realize beam alignment for device characterizations, because other THz components, i.e., wire-grid polarizer (with limited aperture of ~1 inch in diameter) need to be inserted into the measurement setup as shown in Figure S13 in the revised Supplementary Materials.

Figure R1. COMSOL 3D (passive) simulation in half spherical space. (a) 3D perspective view. (b) 2D view cut at $y = 0$ (xz) plane.

Reviewer #1 -- Comment 2:

The power output is rather low. What's the maximum peak power achieved? Which kind of practical use those THz QCLs can have?

Response from Authors:

We thank the reviewer for this comment.

Under the electrical pumping with a pulse width of 500 ns, a repetition rate of 10 kHz, and an additional modulation frequency of 25 Hz for Gentec-EO T-Rad powermeter operation, the maximum output peak power of our single-mode THz vector beam laser is around 1 mW for sample $a = 30 \mu\text{m}$ (vortex core radius $\xi = 2a$, phase modulation radius $4a$) with an effective pumping area of 0.000624 cm^2 . Meanwhile, the ridge lasers fabricated from the same micro-processing batch with much larger device pump area of 0.015 cm^2 (ridge width $100 \mu\text{m}$, ridge length $1500 \mu\text{m}$) deliver a maximum peak power of $\sim 1.54 \text{ mW}$. Therefore, it can be concluded that the output power of THz vector beam laser is mainly limited by the performance of THz QCL wafer used in this work.

The main objective of this work is to demonstrate a novel semiconductor laser cavity design concept, which is capable of emitting desired THz vector beams (again, THz vector beams conventionally are generated by bulky free-space methods which require discrete free-space optical components). The design strategy can be conveniently applied to high-power QCL wafers for high-power emission for practical usages if high performance QCL wafer was utilized. Moreover, such design can also be readily scaled down to shorter wavelengths, such as the mid-infrared, the near-infrared, and the visible frequency regions.

Reviewer #1 -- Comment 3:

The technological impact of those lasers is not discussed at all. What's the advantage they can have over standard THz QCLs?

Response from Authors:

We thank the reviewer for raising this valuable point. As mentioned previously in this response letter, the THz vector beams have various important applications. The special beam profile and polarization properties empower advantages over regular Gaussian beams in high-resolution THz imaging, THz communications, spectroscopy as well as astronomy applications. The existing methods for generating THz vector beams rely on external THz source for optical pumping and bulky free-space optical components like half-wave plate, nonlinear crystals, etc. The THz vector beam generator developed in this work is a monolithic semiconductor laser with single mode operation. By simply arranging the designed phase distribution lattices around a cavity vortex core, the topological vortex cavity design strategy in this work is capable of generating the cylindrical vector beam that obtains far-field polarization winding. The standard THz QCLs, such as ridge laser, DFB laser, and traditional photonic crystal laser, have no such flexibility.

To highlight the technical impact of these lasers, we have revised the manuscript in the “Discussion and Conclusion” section (on pages 5 and 6), which reads:

“The CV beams, with their doughnut-shaped beam profiles that can give rise to tightly focused spot, have found revolutionary breakthroughs in super-resolved fluorescence microscopy [37-39]. They also carry different polarizations, such as radial polarization, azimuthal polarization, and spiral polarization (the current study shows such a polarization), that can be applied to different types of super-resolved microscopies, optical trapping and manipulation, laser machining, and charged particle acceleration. These advanced applications have been reviewed by Ref. 40. THz wave emission devices are least developed, compared to the wave generators in the adjacent infrared and microwave frequency ranges. The realization of THz CV beams through the proposed integrated approach can find its significance for applications in high-bandwidth wireless communications, THz security inspection, THz spectroscopy, bio-imaging, etc. The demonstrated THz CVB source is monolithically integrated and electrically pumped, which can be easily integrated onto photonic chips for applications listed above. Last but not least, our laser cavity design only relies on the dielectric refractive index modulation. Therefore, it can be easily scaled to other wavelength regimes, such as the mid-IR, the near-IR, and the visible regions.

In summary, we have demonstrated the first electrically pumped THz lasers based on photonic Majorana zero modes, which generate cylindrical vector beams with nonzero polarization winding. The laser emission is dominated by a mid-gap Majorana zero mode, with a SMSR of >14 dB. The winding of the CV beam is intrinsically tied to the vorticity of the Kekulé modulation on a photonic lattice, demonstrating how 2D topological modes can affect the topological features of the laser emission in the far field. It is worth mentioning that the laser cavity design in this work enables control of emission spectrum and far-field beam polarizations simultaneously, making Majorana-like photonic QCLs unique as compared to standard THz QCLs, such as ridge laser, distributed-feedback laser, and traditional photonic crystal laser which have no such flexibility. The emission properties such as output powers, beam divergence, polarizations can be further optimized through tailoring the structural parameters such as air hole radius and cavity size.”

Reviewer #1 – Comment 4:

How can the author exclude contribution coming from disordered modes, like the ones arising in a random laser? Can the authors comment on that?

Response from Authors:

We thank the reviewer for raising this valuable point. To clarify the role of disordered modes, we would like to make a comparison between the emission properties of a random laser and those of the topological laser in this work.

Photonic structures consisting of disorderedly arranged units can support random modes with irregular mode profiles localized to certain spatial regions, due to the multiple scattering of light and the Anderson localization effect. Random lasing typically has the following four characteristics. 1) Due to the irregular mode profiles and the corresponding random frequencies, disordered laser systems are commonly featured with multiple random lasing peaks, which has been reported in several representative THz quantum cascade random lasers (*Optica*, 2016, 3(10): 1035-1038; *Light: Science & Applications*, 2019, 8(1): 1-13; *Nature communications*, 2020, 11(1): 1-8; *APL Photonics*, 2020, 5(3): 036102). 2) The multiple emission modes will constructively or destructively interfere in the far field, and thus the farfield pattern of a random laser normally shows irregular intensity profiles. 3) The random laser has complicated lasing dynamics due to its multimode behaviours. Different emission peaks appear (or disappear) when varying the pump intensity (*APL Photonics*, 2020, 5(3): 036102). 4) The lasing spectrum is also sensitive to the external optical perturbation (*Nature Communications*, 2020, 11(1): 1-8) and also the fabrication defects, which normally lead to different lasing spectra from device to device.

In contrast to the disordered photonic structures, the topological cavity utilized in this work has well-arranged air holes designed for certain phase modulation profiles. The cavity hosts a single photonic Majorana zero mode at the frequency located in the mid-gap of the bandgap, as predicted by theoretical calculations. The single emission mode is observed at the center of a large frequency range where no other lasing peaks exist. The large frequency range corresponds to the large photonic bandgap (see Fig. 2a, Fig. 3c and Fig. S3) of the topological photonic structure design. The topological laser in this work shows a well-defined lasing frequency which is stable throughout the whole pumping dynamic range (see Fig. 3) and can be well reproduced by a structure with different lattice parameters (see Figs. S10-S12). The laser also delivers a very clean farfield beam pattern without obvious irregular speckles (see Fig 4). In addition, it can be found the the laser emission spectrum is robust to the pumping condition (e.g. pulse width) and the fabrication defects (see Supplementary Materials Section 7 and Fig. S12). All of these emission features are different from those of a random laser. Thus, it is justifiable to exclude the contributions from disordered modes.

GENERAL COMMENTS FROM Reviewer #1:

The manuscript appears very technical, not conceived for a broad audience but addressed only to researchers working in the field.

For the above reasons I do not recommend publication in Nature Comm. I suggest submitting it to a different journal or transferring to Scientific Reports.

Response from Authors:

We thank the reviewer's for all the constructive comments raised, which helped us to improve our work and the presentation. Here, we would like to take this opportunity to provide further clarifications on the contributions of this work to highlight the fundamental physics underpinning the device design as well as the scientific significance of this work which apply to a wide range of communities in this interdisciplinary research field.

The demonstrated THz QCL in this work relies on the photonic analogy of topological Majorana zero mode (i.e. isolated mid-gap state), whose concept originates from a chiral edge state circulating

a given defect in p-wave superconductors (Nucl. Phys. B 1981, 190, 681). Similar to the edge state of 1D topological insulator (Nat. Photon. 2017, 11, 651-656) and corner state of 2D and 3D topological insulators (Science 2017, 357, 61-66), the topological Majorana zero modes show isolated mid-gap state and tightly localized density of states. Interestingly, they also possess circulating phase from the chiral edge state of Cooper pairs, thereby giving rise to bounded topological vortex for the superconducting physics. We borrow this concept from condensed matter physics and implement it in photonics on a semiconductor laser cavity design for emission characteristic engineering, as a demonstrator in the THz wavelength range. We construct a photonic Majorana-like zero mode by integrating a vortex core and phase modulating lattice around its periphery. The vortex core is given by the unmodulated Dirac-cone photonic lattice, determining the resonant frequency; while the phase accumulation of the resonant mode is achieved by the Kekulé-modulated photonic lattice, with its bandgap opening for modal in-plane confinement. As a result, unlike the nonradiative counterparts of electron/Cooper pairs, the photonic analogy of topological Majorana zero mode can generate cylindrical vector beam emission with polarization singularities in the far field. Such a device can functionize simultaneously as a laser microcavity and a beam engineering element. Therefore, the design principle of Majorana zero mode laser in this work has rich physics background and implications.

In recent years, the concept of topology is no longer limited to condensed matter materials (which was awarded the 2016 Nobel prize in Physics), but has been paradigm-shifted to photonics, mechanics, acoustics, and many other fields. Recent proposals on applying topological concepts in the photonic community have broadened the impact of topology and attracted a lot of interests in photonic, condensed matter physics, and material science communities. For example, the demonstration of topological lasers (Science 2017, 358, 636-640; Science 2018, 359, eaar4005; Science 2018, 359, eaar4003) have brought the photonic branch of topological physics-known as “topological photonics” into active photonic devices. Recently, more and more new physics have been investigated in topological lasers under optical pumping, such as spin-momentum locking to emit vortex laser (Nat. Phys. 2021, 17, 700-703), artificial long-range interaction to mimic magnet-free Haldane model (Nat. Phys. 2021, 17, 704-709), generation of higher-dimensional supersymmetry to enhance the single-mode laser emission with lower divergence angle (Science 2021, 372, 403-408), and so on. The demonstrated photonic Majorana laser is among such initiatives in this emerging interdisciplinary research field which achieves well control of emission spectrum and far-field polarizations.

The demonstration of monolithic THz vector beam generator will attract a lot of interests from different research fields. The laser photonic design principle in this work paves the way for laser beam/polarization engineering while maintaining single-mode operation. Majorana zero mode is a new type of topological protected states, which endows topological laser a compact footprint. The application of Majorana zero mode for electrically pumped laser emission with single mode operation pushes the research of photonic Majorana zero mode as well as topological lasers one step further for real applications. The Kekulé phase modulation of the photonic lattice relates the near-field phase to far-field polarizations and beam profiles while still maintaining the in-plane resonances. It borrows the idea of spatial phase modulation by using a metasurface, but more suitable to be applied in on-chip resonant photonic devices such as laser, resonator and filter. Basically, our laser cavity design only relies on the dielectric refractive index modulation. Therefore, it can be easily scaled to other wavelength regimes, such as the mid-IR, the near-IR, and the visible regions, which could be developed for more potential applications. Last but not least, the THz vector beam has many important applications as mentioned in the previous replies in this response letter.

In summary, our demonstrated THz vector beam laser that is based on innovations from the physical mechanism, photonics design, and laser implementation. It shows great potentials for many applications in a broad frequency range. It will appeal to a broad audience from diversified research communities including but not limited to condense matter physics, laser physics and engineering, photonic crystals, metasurfaces, THz, infrared, and visible applications. We hope that the reviewer will be convinced by the revised manuscript and our responses to the comments.

GENERAL COMMENTS FROM REVIEWER #2:

The authors demonstrated an electrically pumped terahertz topological laser. The laser emission is dominated by mid-gap Majorana zero modes, which form cylindrical vector (CV) beams with nonzero polarization winding. The winding of the CV beam is intrinsically tied to the vorticity of the Kekulé modulation on the photonic lattice, which was achieved by creating air holes through the semiconductor active region and top metal contact. The device exhibit good performance with peak power over 1 mW, threshold current density of around 1.6 kA/cm², and side-mode-suppression-ratio (SMSR) of more than 16 dB over a single weak high band edge mode. The experimental data is well explained and confirmed by the theoretical simulation, demonstrating the first photonic Majorana quantum cascade laser, which can have important applications in wireless communications, terahertz microscopy and imaging. The conclusions are supported by the experimental and theoretical results. All Figures are well plotted and the manuscript (including the supplementary materials section) is well drafted. I would recommend to publish this paper.

Response from Authors:

We thank the reviewer for the supportive and encouraging comments, especially that our work “exhibit good performance”. Below, we will respond to the reviewer’s specific comments one by one.

DETAILED TECHNICAL COMMENTS FROM REVIEWER #2:

Reviewer #2 -- Comment 1:

For those theoretical simulations/experimental measurements in Figures 3 and 4, and Figure S8 in the sections 5 and 6 and Figure S12 in the supplementary materials, at what temperature was the device simulated/measured?

Response from Authors:

The theoretical simulations and experimental measurements across the whole project were conducted at about 8.5 K. Before the THz topological laser device design, conventional ridge lasers were fabricated and characterized for material effective refractive index and THz quantum cascade laser (QCL) wafer performance estimation. The characterization of THz ridge lasers was performed in a helium-gap-steam cryostat with temperature stabilized at ~8.5 K and driven by an electrical pulse generator with repetition rate of 10 kHz. The emission spectra across the whole pump dynamic range indicated the gain spectral range and the gain center, the pump dynamic range provided important information of lasing threshold and rollover current density (also voltage), and the emission power helped us to estimate the possibility for far field measurement. And most importantly, the free spectra range (FSR) of emission spectra helped us to estimate the effective refractive index of the QCL active region to be around 3.85 at the operation frequency, which was then employed to locate the photonic Majorana zero mode frequency. The fabricated topological cavity lasers were also characterized at the same temperature.

The reason for conducting all the simulations and measurements under low temperature is due to the fact that THz QCLs now cannot be operated at room temperature. A new maximum operating temperature record has been achieved recently at 225 K (Nature Photonics, 2021, 15(1): 16-20), but so far most THz QCLs can only operate at low temperature. The THz laser transition happens between two intersubband energy levels with very small energy difference (~ 20 meV). Influenced by electron-LO-phonon scattering process, temperature broadening of re-absorption transitions, thermal backfilling and electron-electron scattering process, the laser transition will be substantially suppressed when the temperature increases, and thus the gain coefficient reduces. Therefore, it is

very challenging to realize THz QCL operation at higher temperature. The temperature performance of topological cavity laser is limited by the laser gain medium.

Reviewer #2 -- Comment 2:

Has the SMSR been measured at elevated temperatures? Does the SMSR depend on device operating temperature? Does the far-field patterns of the CV beam depend on device operating temperature?

Response from Authors:

We thank the reviewer for raising this valuable point. To further investigate the single-mode operation of our laser device, we measured the temperature-dependant laser spectra, as shown in **Figure R2b**.

Figure R2. Temperature-dependent performance of a photonic Majorana-like single-mode QCL. a, the light-current-voltage (L-I-V) curves at cooling temperatures of 8.5 K, 20 K, and 40 K, respectively. b, the corresponding emission spectra at their roll-over points. c-e, the corresponding far-field profiles. When temperature increases, the signal-to-noise (SNR) of far-field pattern reduces due to weaker output intensities.

To investigate the lasing spectra and far-field emissions at different temperatures, we measured the topological QCL device with +1 winding number, accordingly. **Figure R2a** shows the L-I-V curves at three temperatures, i.e., 8.5 K, 20 K and 40 K, respectively. The I-V curves share the same trends when the external pump increases. This means that the conductance of QCL gain medium and contacts stays almost unchanged. As mentioned previously, when the ambient temperature increases, the gain coefficient of QCL wafer will reduce due to many possible carrier processes. Therefore, the light intensities decrease as expected when temperature ramps from 8.5 K to 40 K. The SMSR is also slightly degraded from 14.7 dB for 8.5 K to 10.6 dB for 40 K, but the Majorana lasing mode is pinned to a frequency and always dominant the emission process. Most importantly, the emitted beams measured at these temperatures always show doughnut shapes as shown in **Figures R2c-2e**, with gradually degraded signal-to-noise (SNR) when temperature rises. Such a

phenomenon, the fixed emission frequencies and consistent beam profiles, in some sense, demonstrates the topological robustness of the Majorana-like photonic QCLs that not only for the lasing frequency, but also for the emitted far-field beam profiles.

As an important performance supplement, we discuss the device temperature performance and put these results in the revised supplementary Figure S4 on page 5.

Reviewer #2 -- Comment 3:

The Figure S8 in the section 6 of the supplementary materials should be Figure S9. The onward figure numbers should be changed accordingly.

Response from Authors:

We thank the reviewer for pointing out this error. It has been revised accordingly in the supplementary material.

Reviewer #2 -- Comment 4:

Reference 30, the title should be “Photonic Topological Mode Bound to a Vortex”, and the year should be (2020).

Response from Authors:

We thank the reviewer for pointing out this mistake. It has been revised accordingly in the manuscript.

Reviewer #2 -- Comment 5:

If possible, can the authors include the active region design information of the QCL in the supplementary materials section?

Response from Authors:

We thank the reviewer’s constructive comments. The THz QCL active region design and band diagrams have been included in the Supplementary Materials Section 1. We also include the QCL active region here, as shown in **Figure R3**. Usually, laser emission from a QCL is achieved through electron intersubband transitions in a repeated stack of semiconductor multiple quantum well heterostructures. Here, “cascade” refers to the movement of electrons which transit from an upper state (u) to a lower state (l), relax quickly to the injector band (inj) through electron-LO-phonon scattering process and then tunnel “horizontally” to the upper state (u) of the next period of the stack. The “QCL” name is tied to the gain medium and is not related to any specific cavity design. The laser cavity could be a microdisk, a Bragg cavity, a photonic crystal, or a photonic crystal with topological cavity.

Figure R3. Simplified conduction band diagrams and the moduli squared of wavefunctions of the THz QCL active region design used in this work. The active region consisted of 228 module repeats with alternating barriers ($\text{Al}_{0.15}\text{Ga}_{0.85}\text{As}$) and quantum wells (GaAs). Starting from leftmost barrier of each period, the layer thicknesses in nm (with barriers indicated in bold-face font) are 8.15/**2.46**/8.9/**4.3**/16/**4.1** and the widest well is n-doped with Si at $7 \times 10^{16} \text{ cm}^{-3}$ in the central 5 nm region. The bias voltage is 59 mV/module.

Reviewers' comments:

Reviewer #1 (Remarks to the Author):

The authors convincingly addressed the points I have raised

The methodology is sound, the work solid and claims and conclusions carefully supported

I still found this work very technical and not for the broad audience of readers that a journal as Nature Communications should embrace.

Reviewer #2 (Remarks to the Author):

The authors have addressed all my concerns/comments. I believe the manuscript presents novel results, which should justify its publication in Nature Communications. I would recommend to publish it as is.

Response Letter to the Reviewers – 2nd round review

Thank you again for your kind help on handling our manuscript (NCOMMS-22-05649A), “Photonic Majorana Quantum Cascade Laser with Polarization-Winding Emission”. We also appreciate the efforts of the two referees who reviewed the manuscript. We sincerely appreciate the full supports from the Reviewer #2 who recommended to publish it as it is and the positive comments by the Reviewer #1 on “The authors convincingly addressed the points I have raised. The methodology is sound, the work solid and claims and conclusions carefully supported.”

In the response letter below, each reviewer comment is quoted in italics and is followed by the corresponding detailed response. In the text below, changes to the revised manuscript as a response to the reviewers’ comments are highlighted in blue color and replies/clarifications regarding the reviewer’s comments are also provided accordingly.

GENERAL COMMENTS FROM REVIEWER #1:

The authors convincingly addressed the points I have raised. The methodology is sound, the work solid and claims and conclusions carefully supported. I still found this work very technical and not for the broad audience of readers that a journal as Nature Communications should embrace.

Responses by Authors:

We sincerely appreciate for the Reviewer #1’s time and efforts in reviewing this manuscript, and the positive comments “The authors convincingly addressed the points I have raised. The methodology is sound, the work solid and claims and conclusions carefully supported.”, but we are very puzzled by the Reviewer #1’s conclusion on “I still found this work very technical and not for the broad audience of readers that a journal as Nature Communications should embrace.” which, in our opinions, is very vague and subjective.

1. If the concern on “very technical” means “very engineering in development”, we would like to clarify that our demonstrated single-mode THz vector beam emitting laser in fact stems from physical concepts in condense matter physics, i.e., topological Majorana zero mode which originates from a chiral edge state circulating a given defect in p-wave superconductors (Nucl. Phys. B 1981, 190, 681). The topological Majorana zero mode shows an isolated mid-gap state with tightly localized density of states, and more interestingly, with a vortex phase from the chiral edge state of Cooper pairs (Nat. Rev. Mater. 2021, 6, 944-958). We implemented such Majorana zero mode concept in photonics on a THz quantum cascade laser (QCL) platform by carefully designing a cavity core and phase modulating lattices around its periphery. Unlike the counterparts of electron/Cooper pairs in superconductors which are nonradiative, the photonic analog can generate single-mode emission due to the mid-gap state nature and cylindrical vector beam emission (including azimuthal and radial polarizations) owing to the phase modulation property. Thus, the design of photonic Majorana zero mode laser has rich physics background and implications.

2. If the concern is on “not for the broad audience of readers that a journal as Nature Communications should embrace”, we would like to justify that in our opinion the demonstrated results will attract interests from various research fields. **(1)** For laser communities, our laser cavity design which is used to achieve beam/polarization engineering while maintaining single-mode operation just relies on dielectric refractive index modulation, thus the concept can be easily extended to other wavelength regimes, e.g., the mid-IR, the near-IR, and the visible regions, beyond the THz regime as demonstrated in this work. **(2)** In addition, the demonstrated research falls into an emerging interdisciplinary research area of topological physics which is not only relevant to photonics or condensed matter materials but also extended to mechanics, acoustics, thermal engineering, and other diversified fields. **(3)** Furthermore, the Kekule phase modulation of the photonic lattice demonstrated in this work relates near-field phase to far-field polarizations and beam profiles, which is analogous to spatial phase modulation by using a metasurface which could have impacts in the metasurface community, i.e. rather than utilizing photonic lattice structures as we demonstrated in our work, one can also explore using metasurfaces. **(4)** Last but not least, the demonstrated THz laser itself can achieve controlled vector polarization beam while maintain single-mode lasing over the entire dynamic range of the laser in a monolithically integrated chip, which is unique as compared to the existing THz QCLs. We have shown the comparison of traditional THz QCLs with metasurface/plasmonic designs and our proposed photonic Majorana zero mode design as shown in Table 1 below. Our demonstrated laser with cylindrical vector beam and single-mode operation has many potential applications in THz imaging, communications, spectroscopy, astronomy, and charged particle acceleration, which are of great interests to the THz photonics community. Thus, the demonstrated results appeal to a broad audience in diversified research communities under Nature Communications.

3. We would also like to elaborate on the **three main impacts on technology developments** of our work:

- **For the impacts in the THz photonics, particularly THz QCLs**, THz single-mode lasers with vector beam emissions find important applications in various fields as shown in Figure 1 below. Traditional approaches for generating THz vector beams with single-mode emissions use free-space optical components like wave plates, polarizers, and nonlinear crystals, etc. However, they are bulky, expensive, and sensitive to environmental vibrations which leads to packaging issues. Several research groups started to explore the possibility of integrating plasmonic/metasurface structures onto a THz QCL for polarization controls. However, all these approaches can only realize linear polarized and circular polarized beams and none of these can achieve more complicated cylindrical vector beams (including both radial and azimuthal polarizations) emissions with a nontrivial polarization winding, while simultaneously achieving single-mode emission. **In our work, we utilize a novel integrated design based on topological photonic Majorana zero mode (MZM) designs to achieve the first electrically-pumped THz laser with cylindrical vector beam emission and single-mode operation over the entire dynamic range in one monolithically integrated**

laser cavity, simultaneously. The polarization singularity of the far-field beam and vector beam emission stem from the Kekule phase modulation of near-field vectorial electromagnetic fields surrounding a laser core. Meanwhile, the laser core works as a compact resonator to support single mode operation. The polarization winding number of vector beam can be further engineered by modulating the phase distribution of topological lattice. We would like to highlight that our design just relies on dielectric refractive index modulations, thus the concept can also be easily extended to other wavelength regimes, e.g., the mid-IR, the near-IR, and the visible regions, beyond the THz regime as demonstrated in this work.

Comparison	Polarization control	Advanced applications enabled by Cylindrical Vector (CV) beams
Previous designs	Linear and circular polarizations achieved by metasurfaces and plasmonics designs	N.A.
Our work	Cylindrical vector beams with radial and azimuthal polarizations controlled by winding of the Majorana zero modes	 1. Tight focusing [Opt. Lett. 32, 2535-2537(2007)]. 2. Particle manipulation [Appl. Opt. 48, 6143-51 (2009), & Nat. Commun. 3, 998 (2012)]. 3. Moving particle inspection [Optica 2, 1-5 (2015)]. 4. Super-resolved fluorescence microscopy [Opt. Lett. 19, 780-2 (1994); Science 316, 1153-8 (2007); Opt. Express 12 3605-17 (2007)]. 5. Communications [Opt. Lett. 40 4843-6 (2015)]. 6. Quantum optics [Nat. Commun. 6, 7706 (2015)].

Figure 1. Comparison of THz QCLs with metasurface/plasmonic designs and the demonstrated photonic Majorana zero mode design with Cylindrical Vector (CV) beam in this work.

- **For impact in topological photonics, particularly topological lasers,** our photonic Majorana quantum cascade laser is the first electrically-pumped single-mode topological laser with a compact footprint. The development of electrically pumped topological insulator lasers has drawn a lot of research interests recently because it is considered as one of the most promising directions with great prospects in pushing the boundaries of topological photonics

into practical applications. So far, only two works, i.e., Nature 578, 246-250 (2020) (our previous work) and Nat. Comm. 12, 3434 (2021) have been reported on realizing electrically-pumped topological lasers. However, they have obvious limitations as summarized in Table 1, i.e. the Nature 578, 246-250 (2020) work operates in multi-mode lasing and the Nat. Comm. 12, 3434 (2021) work has large ($\sim 212 * \lambda_{laser}$) laser footprint. Another limitation of these two lasers is that they are not capable of generating any specially polarized beams which is limited by the fundamental physics behind their designs, i.e. Valley-Hall photonic design and quantum spin Hall photonic design, respectively. **Our work demonstrates a compact ($\sim 4 * \lambda_{laser}$) electrically pumped topological laser with single-mode lasing emission over the entire dynamic range in one monolithically integrated laser cavity.** It has a compact size as the resonant mode is localized at the cavity core. It can achieve single mode operation, with a side mode suppression ratio SMSR > 15 dB, since the MZM lasing frequency is away from its competing bulk modes and the net gain is higher than those of other modes when the vertical emission loss is considered. In addition, the demonstrated research has broader impact in topological physics which is not only relevant to photonics but also extended to mechanics, acoustics, thermal engineering, condensed matter materials, and other fields in topological physics.

Table 1. Performance summary of electrically-pumped topological insulator lasers.

Work	Physics mechanism	Device size (* λ_{laser})	Single mode	Polarization control	Selective pump
Nature 2020	Valley-Hall edge state design	~ 9	No	No	Yes
Nat Comm 2021	Quantum spin Hall edge state design	~ 212	Yes	No	Yes
Current work	Majorana zero mode design	~ 4	Yes	Yes (Cylindrical vector beam)	No

- **For impact on laser engineering:** We would also like to highlight another unique lasing property of our laser is that it has great promise in achieving high-power high-brightness single-mode operation due to its mid-gap lasing mode nature (the lasing mode happens in the middle of the bandgap) which is much more stable than traditional 2-dimensional lasers, e.g. photonic crystal lasers, which rely on band edge modes that can be easily coupled with photonic bulk modes. **Here, we demonstrate this power scaling property with new experimental and simulation results.** We fabricated and measured a device with a larger cavity size (cavity core radius $\xi = 8a$, width of Kekule-modulation part = $9a$, and winding number $w = -1$). For comparison, we also fabricate a ridge laser with similar pump area, we found that the power output of our THz Majorana zero mode topological laser has around 9-fold output power higher efficiency than that of the ridge laser. Our Majorana laser still maintains a very good single-mode lasing with SMSR > 15 dB, as shown in Figure R2b and R2c. Compared to the MZM topological laser with a cavity core radius of $\xi = 2a$, the light

output power is increased by 8-fold as seen in Figure R2a. Due to the experimental limitation on the pumping current of our laser driver and the limited QCL wafer we have for this work, we could not fabricate larger sizes of MZM topological lasers for experimental demonstration. For further verification on the scalability of our design, we have calculated the cavity eigenmodes with larger cavity core radius using two-dimensional simulations. Although some high-order lateral modes enter the original photonic bandgap when the cavity core becomes larger, the photonic Majorana zero mode is always located at the mid-gap center with a high Q factor (as shown in Figure 2d). It is expected that single-mode operation can be maintained with the cavity core size up to $\xi = 32a$ thanks to its relatively large frequency spacing to the adjacent competing modes and the good overlap of the MZM frequency with the gain peak (~ 3.4 THz) of the QCL active region.

Figure 2. Performance of THz topological MZM laser with a larger cavity size. **A**, Measured L-I-V curve of topological MZM laser with cavity core radius $\xi=8a$, and the overall radius of the laser device is $17a$. Another laser device with cavity core radius $\xi=2a$. The unit cell period a is $30 \mu\text{m}$. **B**, 2D map of the measured lasing spectra at different pump current densities. A stable lasing peak is clearly observed at around 3.4 THz . **C**, The emission spectrum at roll-over point (maximum output power point). The SMSR is larger than 15 dB . **D**, A 2D eigenmode calculations for devices with different MZM core radius (ξ). The color bar shows the Q factor of eigenmodes.

GENERAL COMMENTS FROM REVIEWER #2:

The authors have addressed all my concerns/comments. I believe the manuscript presents novel results, which should justify its publication in Nature Communications. I would recommend publishing it as is.

Response from Authors:

We thank the reviewer for the full supports and recommend publishing it as it is.

Sincerely yours,

Qi Jie Wang, Professor
Fellow of Optica
Associate Chair (Research)
School of Electrical and Electronic Engineering
& School of Physical and Mathematical Sciences
Nanyang Technological University,
Singapore
(On behalf of the other authors of the manuscript)

REVIEWERS' COMMENTS

Reviewer #2 (Remarks to the Author):

Thank the authors for further clarifying the novelty and scientific impacts of their research paper. The experimental and simulation results clearly demonstrate the first electrically-pumped THz laser with cylindrical vector beam emission and single-mode operation over the entire dynamic range, which was achieved through well-engineered topological photonic Majorana zero mode designs. This represents a significant milestone in the development of terahertz quantum cascade lasers. The results in the manuscript, the supplementary information and the response letter have shown convincing impacts in terahertz photonics, topological photonics, and laser engineering. It would be of substantial interest to a broad audience of this journal. I would recommend to publish the manuscript as is.

Response Letter to the Reviewers – 3rd round review

Thank you again for your kind help on handling our manuscript (NCOMMS-22-05649B-Z), “Photonic Majorana Quantum Cascade Laser with Polarization-Winding Emission”. We sincerely appreciate the full supports from the Reviewer #2 who recommended to publish it as it is.

In the response letter below, each reviewer comment is quoted in italics and is followed by the corresponding detailed response.

GENERAL COMMENTS FROM REVIEWER #2:

Thank the authors for further clarifying the novelty and scientific impacts of their research paper. The experimental and simulation results clearly demonstrate the first electrically-pumped THz laser with cylindrical vector beam emission and single-mode operation over the entire dynamic range, which was achieved through well-engineered topological photonic Majorana zero mode designs. This represents a significant milestone in the development of terahertz quantum cascade lasers. The results in the manuscript, the supplementary information and the response letter have shown convincing impacts in terahertz photonics, topological photonics, and laser engineering. It would be of substantial interest to a broad audience of this journal. I would recommend to publish the manuscript as is.

Responses by Authors:

We sincerely appreciate for the full support and highest encourage, which means a lot to us. Indeed, the introduction of topological physics to electrically-pumped laser has shown the capability to overcome the influence of sharp bends and defects on operation mode (Nature 2020). The far-field beam engineering, however, is one of pendent topics not only for the electrically-pumped topological insulator lasers, but also for the other types (photonic crystal, ring resonator cavity) of lasers. Here we believe our demonstration provides one of effective methods for property engineering of THz lasers. Together with the latest breakthrough that allows THz QCL to operate at -12 °C (arXiv:2211.08125), we may witness that the photonic Majorana THz QCLs play pivotal roles in many practical applications.